# Testing the Stable Unit Treatment Variance Assumption (SUTVA) Within Cochrane Reviews of Antimicrobial-Based Pneumonia Prevention Interventions Among Mechanically Ventilated Patients Using Caterpillar Plots

**DOI:** 10.3390/jcm14196841

**Published:** 2025-09-26

**Authors:** James C. Hurley

**Affiliations:** 1Melbourne Medical School, University of Melbourne, Ballarat, VIC 3350, Australia; jamesh@gh.org.au; 2Ballarat Health Services, Grampians Health, Ballarat, VIC 3350, Australia; 3Ballarat Clinical School, Deakin University, Burwood, VIC 3125, Australia

**Keywords:** caterpillar plots, topical antibiotic prophylaxis, chlorhexidine, stable unit variance assumption (SUTVA), intensive care unit

## Abstract

**Background/Objectives**: Causal inference requires validating the stable unit treatment variance assumption (SUTVA). Whilst antimicrobial-based interventions, being topical chlorhexidine and topical antibiotics prophylaxis (TAP), appear effective in preventing ventilator-associated pneumonia (VAP) among ICU patients receiving mechanical ventilation (MV) within randomized concurrent controlled trials (RCCT), SUTVA has never been tested for this inference. **Methods**: Caterpillar plots of the VAP incidence proportions for control and intervention groups within RCCTs obtained from recent Cochrane reviews of antimicrobial-based VAP prevention interventions were derived using random effects methods to enable comparison versus the expert VAP incidence reference range (5 to 40%). **Results**: The summary VAP prevention effect size derived from three categories of 57 RCCTs of antimicrobial-based interventions was recapitulated. The VAP incidences of 24 control groups and 6 intervention group incidences were above, whereas only 1 and 6, respectively, were below the expert VAP incidence reference range (5 to 40%) (*p* < 0.001; chi-square = 17.42; df = 2). The results after excluding 18 low-quality studies were similar. Paradoxically, the 90% prediction limits in association with the summary control group incidences are each approximately 20 percentage points wider than for those associated with the intervention group summaries. **Conclusions**: Control group VAP incidences above and more dispersed versus the expert opinion VAP range are common within the Cochrane reviews of antimicrobial-based VAP prevention interventions. Recognition requires an arms-based analysis using caterpillar plots. The SUTVAs that underlie the inference of prevention from the effect size estimates are not valid.

## 1. Introduction

Interventions based on the use of topical antimicrobials, being either topical antibiotic prophylaxis (TAP) or topical chlorhexidine, toward preventing ventilator-associated pneumonia (VAP) have been widely studied within randomized concurrent controlled trials over the past 40 years [1,2,3,4,5,6,7,8,9,10,11,12,13,14,15,16,17,18,19,20,21,22,23,24,25,26,27,28,29,30,31,32,33,34,35,36,37,38,39,40,41,42,43,44,45,46,47,48,49,50,51,52,53,54,55,56,57,58,59,60,61,62].

The most recent systematic reviews of the randomized concurrent controlled trials of these interventions used as prevention within populations of patients receiving prolonged mechanical ventilation suggest that they appear highly effective. Moreover, the study data on which this summary VAP prevention evidence is based are generally rated as either ‘high quality’ or ‘high-certainty evidence’ for both types of antimicrobial intervention [59,62].

The Stable Unit Treatment Variable Assumption (SUTVA) is required for valid inference from effect size estimates derived from randomized concurrent controlled trials and, by extension, from the summary effect size estimates within systematic reviews [63,64]. The SUTVA has two elements, consistency and spillover. However, testing these elements is not as simple as it requires reference to an external benchmark [65]. Neither element is considered within the scoring of study quality. In any case, the SUTVA elements can generally not be tested within any given randomized concurrent controlled trial or systematic review.

Several published potential range estimates are available against which to benchmark the incidence of VAP amongst the population of ICU patients receiving prolonged mechanical ventilation in the literature. For example, the most recently published systematic review states this incidence to be 5 to 40% of patients receiving prolonged mechanical ventilation [62].

The objective here is to test the SUTVA amongst the findings of Cochrane reviews of antimicrobial-based VAP prevention interventions [58,59,60,61,62]. This requires an arms-based analysis of the VAP incidence proportions within the control and intervention groups of these randomized concurrent controlled trials versus the most recently stated expert range.

## 2. Materials and Methods

### 2.1. Literature Search and Study Decant

This analysis is based on a previous search of the Cochrane database for systematic reviews of interventions studied with randomized concurrent controlled trials for the prevention of VAP among mechanically ventilated patients in the ICU [66,67]. From this search, five Cochrane reviews of randomized concurrent controlled trials of topical antimicrobial-based interventions were identified for which VAP counts were abstracted. The VAP counts in previous versions of each systematic review were compared with the most recent [58,59,60,61,62]. The categorization of randomized concurrent controlled trials according to intervention follows that in the most recent Cochrane reviews [59,62]. Note that a duplex randomized concurrent controlled trial is one in which the control group received protocolized parenteral antibiotic prophylaxis (PPAP).

The following randomized concurrent controlled trials were excluded: randomized concurrent controlled trials with fewer than 20 patients in total, and randomized concurrent controlled trials for which only per-protocol data were available. This being a review of the SUTVA within the systematic reviews, a search for randomized concurrent controlled trials in addition to those included in the systematic reviews was not undertaken. Also, given there is no possibility for spillover to occur within randomized concurrent controlled trials with a non-concurrent design, any studies with a non-concurrent design were excluded.

Ethics committee review of this study was not required as this is an analysis of published work, and each individual included study had been approved by an ethics committee.

The data was abstracted as intention-to-treat (ITT) data. Several discrepancies were identified in the VAP counts abstracted within the two most recent systematic reviews of TAP, which appear to relate to whether the data was abstracted from the study publications as per-protocol (PP) data, as in the most recent TAP ± PPAP systematic review [62], versus intention to treat (ITT), obtained by personal communication from the study authors, as in earlier reviews of the TAP ± PPAP randomized concurrent controlled trials [60,61]. There were trivial discrepancies between the two most recent Cochrane reviews of topical Chlorhexidine, which both provide ITT data for all except one randomized concurrent controlled trial [58,59].

The study quality scoring for the various domains, including those such as sequence generation and allocation concealment (selection bias), blinding of participants and providers (performance bias), blinding of outcome assessor (detection bias), incomplete outcome data (attrition bias), and selective outcome reporting (reporting bias) as scored in each Cochrane review, was extracted. As these were scored using a variable number of score elements in each systematic review, here, these are harmonized by conversion to a simple majority or minority score of the various quality elements, as previously harmonized [67].

### 2.2. Contrast-Based Analysis-Method

The study-specific and overall summary VAP prevention effect sizes and associated 95% confidence intervals were calculated for each category using random-effect methods of meta-analysis. The data was set for two group comparisons of binary outcomes using the ‘meta’ command in Stata 18 (Stata Corp., College Station, TX, USA) using the default restricted maximum likelihood [68].

### 2.3. Arms-Based Analysis and Caterpillar Plots

The data from the component control and intervention groups of randomized concurrent controlled trials were decanted from each randomized concurrent controlled trial with care to include the groups from the three-arm studies only once. The summary VAP incidence proportion data was derived using the ‘meta’ command in Stata 18 and the default restricted maximum likelihood. Caterpillar plots, being a forest plot with the group ordered by increasing the VAP incidence proportion, were derived [69]. The concept of a caterpillar plot and the use of benchmarks are illustrated in Figure 1. The associated 95% confidence interval, I^2^ heterogeneity estimates, and 90% prediction limits were calculated using random-effect methods of meta-analysis.

## 3. Results

### 3.1. Characteristics of the Randomized Concurrent Controlled Trials

There were 57 published randomized concurrent controlled trials identified by the search (Table 1, Table 2 and Table 3). One randomized concurrent controlled trial with fewer than 20 patients in total [56], one study with a non-concurrent design [57], and one randomized concurrent controlled trial with only per-protocol data available [17] were excluded.

The randomized concurrent controlled trials were streamed into three categories.

Three randomized concurrent controlled trials provide data for a second intervention arm, and data missing for the third arm from two randomized concurrent controlled trials was recovered from the original sources.

Most randomized concurrent controlled trials originated from ICUs in Northern Europe (n = 21), Southern Europe (n = 13), North America (n = 4), or South America (n = 4) (Figure 2). There were twelve randomized concurrent controlled trials from trauma ICUs, one from a cardio-thoracic ICU, and two randomized concurrent controlled trials from pediatric ICUs.

The randomized concurrent controlled trials were published between 1987 and 2019, with the interquartile range (IQR) being 1998–2010. The TAP and TAP + PPAP randomized concurrent controlled trials were mostly published in the late 1990s, and the topical chlorhexidine randomized concurrent controlled trials were mostly published two or three decades later. A majority quality score rating was achieved for 16 of 18 randomized concurrent controlled trials of TAP + PPAP versus 28 of 39 randomized concurrent controlled trials in the other 2 randomized concurrent controlled trial categories.

Among the TAP duplex and TAP + PPAP categories, there were at least 14 different TAP components within the study intervention regimens. There are also at least 6 different PPAP components within the study intervention regimens.

### 3.2. Contrast-Based Analysis

The summary effect sizes implied all three randomized concurrent controlled trial categories of intervention prevented pneumonia, although there was inequality in the summary effect sizes, with the TAP + PPAP category showing the strongest apparent prevention effect size (Table 4). The associated I^2^ measures of heterogeneity were all in the range of 60 to 70%.

### 3.3. Arms-Based Analysis

There were three indicators that there was more dispersion amongst the VAP incidences of the control versus intervention groups. This is visually apparent among the caterpillar plots (Figure 3). Second, the increased dispersion is also apparent in the 90% prediction limits, which were approximately 20 percentage points wider, in association with the control group versus intervention group summary incidences within each of the three categories of randomized concurrent controlled trial. Thirdly, the I^2^ was up to 5 percentage points higher in association with the control group summary VAP incidences than the intervention group summary VAP incidences. The results after excluding 18 low-quality randomized concurrent controlled trials were similar.

The VAP incidences of 24 control groups and 6 intervention groups were above the expert VAP incidence reference range (5 to 40%), whereas only 6 of 56 intervention groups across the 3 categories were below 5% (*p* < 0.001; chi-square = 17.42; df = 2) (Figure 3).


## 4. Discussion

The Stable Unit Treatment Variable Assumption (SUTVA) is required for valid inference from effect size estimates derived from randomized concurrent controlled trials. By extension, SUTVA is also required for valid inference from systematic review summary effect size estimates. The SUTVA has two elements, consistency and spillover.

In general, systematic reviews include high-quality randomized concurrent controlled trials with the aim that the summary effect size estimates derived are robust and unconfounded. Of note, neither element of SUTVA is considered within the scoring of study quality. Moreover, testing these elements requires an arms-based analysis of event rates together with reference to benchmarks external to the specific randomized concurrent controlled trial or systematic review in question. Benchmarking is rarely undertaken within a systematic review, even though a benchmark range together with a citation source might be mentioned in the systematic review introduction. Most typically, the analysis is limited to a contrast-based analysis by which the effect size estimates are derived without an arms-based analysis of event rates.

Spillover effects have never been considered within Cochrane reviews of topical antimicrobial interventions for preventing infections within ICU populations. Valid inference from randomized concurrent controlled trials presumes there is no spillover. Given, on the one hand, these interventions appear highly effective in preventing both VAP and mortality, whereas on the other hand, patients acquire infections while in the ICU from the ICU environment, the possibility that these interventions might have spillover effects warrant careful consideration, especially given their candidature as population-based interventions, such as in the ‘pneumonia zero” initiative [70].

The use of caterpillar plots is a novel approach to examining the incidence of the event rate of interest amongst the respective control and intervention arms of a randomized concurrent controlled trial within a systematic review [69]. The caterpillar plot is essentially a forest plot with the studies arrayed in order of increasing VAP incidence, together with external benchmarks. This allows a visual inspection of the pattern and a more robust means of identifying potential outlier studies that might disproportionately influence the summary result. Of note, a typical systematic review will have the studies arrayed by alphabetical order of the first author’s name. This traditional array is wasteful of study information as it fails to adequately characterize dispersion in the visual display.

### 4.1. Choice of Benchmark

The benchmark used here is the VAP incidence range as appearing in the most recently published systematic review [62]. There are several incidence range estimates that were available for selection (Table 5), and these have been derived from various original sources. The one chosen for use here is neither overly conservative nor liberal with respect to VAP incidence limits and can be considered a consensus benchmark amongst those that have been provided within the systematic reviews that were the source for the randomized concurrent controlled trials used in this analysis.

Other benchmarks for VAP incidence that could have been used were substantially more conservative (i.e., narrower). For example, a widely cited narrative review for VAP [80] with over 4300 citations in Google Scholar [as of August 2025] gives an incidence range of 8 to 28%.

Another potential benchmark is one derived from an international survey of 1873 patients receiving mechanical ventilation for > 48 h in eleven countries. In this survey, the VAP incidence was 15.6% globally. Across sites in the United States, Latin America, the Asia-Pacific, and Europe, the incidences were 13.5%, 13.8%, 16%, and 19%, respectively [86]. Another potential benchmark is one derived from 109 randomized concurrent controlled trials of non-antimicrobial methods of VAP prevention. From these 109 randomized concurrent controlled trials, the summary control group incidence was 23% (95% CI 20–26%; 95% prediction interval 5.5–52%; I^2^ 86.5%) and the summary intervention group incidence was 19% (95% CI 17–21%; 95% prediction interval 5.1–52%; I^2^ 82.9%) [87].

Regardless of which benchmark range might have been chosen, surprisingly, the summary 90% confidence or prediction limits derived here for each of the three categories of intervention groups are each more closely aligned with these expert range incidences than the 90% confidence or prediction limits derived from the control groups’ summary incidence.

The vagaries in defining and documenting the VAP endpoint have previously drawn comment. VAP is considered a somewhat subjective endpoint with the potential to allow the introduction of bias in low-quality randomized concurrent controlled trials without observer blinding. The basis for this suspicion arises from a previous analysis of 32 TAP ± PPAP randomized concurrent controlled trials, which found that as the certainty of study blinding and other study design quality parameters increased, the effect size for VAP prevention decreased, and the variance of the apparent TAP ± PPAP effect size increased [88]. This observation may also have resulted in the choice [62] of per-protocol data as appearing in the published version of the original randomized concurrent controlled trials versus the ITT data as had appeared within earlier versions of Cochrane reviews of the TAP randomized concurrent controlled trials [60,61].

### 4.2. SUTVA: Consistency Element

The consistency element of SUTVA is that the intervention is consistently received by the individuals assigned to receive it. For example, this element would be violated if some recipients within the intervention group of a randomized concurrent controlled trial received a dose of the intervention that was double that received by other individuals in the intervention group. This would also be violated if some randomized concurrent controlled trials within a systematic review investigated a double dose in comparison to other randomized concurrent controlled trials. This element would also be violated if there was uneven spillover or contagion of the infection of interest to the control group from other individuals within the study population, or an uneven spillover more apparent for groups of some randomized concurrent controlled trials in comparison to others within a systematic review.

This element has not been specifically considered here. However, there are at least a dozen different TAP regimens among the TAP randomized concurrent controlled trials alone. Moreover, for the consistency element, this would need to consider in addition factors such as dosage, route, and timing of administration of these interventions. Moreover, the timing of cessation of these interventions, which is often poorly documented within these randomized concurrent controlled trials, is also a major consideration for their use within ICU populations, given the potential for rebound colonization following the cessation of antimicrobial decontamination.

These factors would suggest that the consistency element of the SUTVA is invalid here. Any consideration of consistency for the TAP randomized concurrent controlled trials would require an appeal to a presumption that the TAP regimens achieve a state of ‘colonization resistance’ that is thought to be associated with achieving a state of ‘selective digestive decontamination’ as originally conceived. This has never been demonstrated.

The consistency element in relation to the use of topical chlorhexidine is somewhat less problematic, but even within the topical chlorhexidine randomized concurrent controlled trials, there are applications of topical chlorhexidine as gel and solution applied with or without the concomitant use of toothbrushing.

### 4.3. SUTVA: Spillover Element

The spillover element of SUTVA, that the treatment received by one individual should not affect the event of interest in other individuals, was anticipated in two postulates stated in the first study [89], being postulated as “*…having heavily contaminated patients next to decontaminated patients might adversely affect the potentially beneficial results* [postulate one]. *Secondly, a reduction of the number of contagious patients by applying [selective digestive decontamination] SDD in half of them, might reduce the acquisition, colonisation and infection incidence in the not-SDD-treated control group* [postulate two]” [89]. That antimicrobial interventions might mediate a population level of infection prevention within the ICU through a mechanism that alters the ICU microbiome to achieve colonization resistance [90] has never been demonstrated, although given the herd protection that occurs with some population-based vaccination interventions, this population effect of antibiotics used as prevention might appear plausible. However, there have been no attempts to test either postulate in the subsequent four decades.

Whilst the observed event rates on which the summary effect size estimates were derived are likely to be robust within high-quality studies, it remains possible that the event rate for the whole randomized concurrent controlled trial population (i.e., concurrent control and intervention groups) may be elevated by the antimicrobial study intervention as a result of rebound and spillover. Hence, there may be dual effects operating, an individual-level effect and a population effect.

### 4.4. Limitations

The main limitation to be considered is the question of randomized concurrent controlled trial inclusions. This being a review of the SUTVA within the systematic reviews of antimicrobial-based interventions, a search for randomized concurrent controlled trials not included in the systematic reviews identified by the search was not undertaken. Moreover, the randomized concurrent controlled trials were identified for inclusion from across more than one systematic review in the case of each of the three categories of intervention to avoid limiting the focus to a single systematic review.

Four large cluster randomized trials of each of TAP [57,91,92] and topical chlorhexidine [93] used for the prevention of acquired infection within this patient population were not included here. Only one of these large cluster randomized trials [57] was abstracted in one of the systematic reviews, and, in any case, none provided data for the VAP endpoint. These four cluster randomized trials are notable for two reasons. Firstly, they each included between 3000 and 9000 patients from at least 13 ICUs in each cluster randomized trial, a considerably larger number of patients than the total number included in all of the randomized concurrent controlled trials within the systematic reviews. Second, their results for the mortality endpoint differ strikingly from the summary estimate derived for the randomized concurrent controlled trials. For the randomized concurrent controlled trials, there is a significant 15-percentage point ICU mortality difference between control versus TAP intervention groups derived from 27 randomized concurrent controlled trials, versus a zero-percentage point ICU mortality difference between control versus TAP intervention groups derived from 3 large cluster randomized trials [94]. This difference between the results of cluster randomized trials versus randomized concurrent controlled trials is contrary to the direction of difference that would be expected if spillover from the TAP intervention had a favorable population effect in preventing ICU mortality among patients concurrent within the same ICU, as originally postulated [89].

### 4.5. Strengths

The caterpillar plot is a novel and simple method to enable the incorporation of external data, which, here, is the VAP benchmarks (Table 5), into systematic reviews. These plots enable a visual assessment of the SUTVAs that are crucial to inferences arising from any analysis.

Of note, the summary effect size estimates obtained here for the three broad categories of intervention are comparable, allowing for differences in the numbers of randomized concurrent controlled trials included, the specific methods of analysis (e.g., Mantel–Haenszel versus restricted maximum likelihood) and summary statistics used (e.g., risk ratios versus odds ratios), versus the five respective systematic reviews [58,59,60,61,62] from which the randomized concurrent controlled trials are drawn from.

Through including a broad selection of randomized concurrent controlled trials from within several systematic reviews, it is possible to state that the unusually high VAP incidence among the three categories of randomized concurrent controlled trials is not attributable simply to the per-protocol nature of the data, as listed in the most recent TAP systematic review [62].

### 4.6. Comparison of Findings

The effect size estimates here are comparable to those estimated in the source systematic reviews [58,59,60,61,62], even though the inferences here diverge from those in the sources. With respect to chlorhexidine mouthrinse or gel, the first systematic review (18 randomized concurrent controlled trials; 2451 participants) estimated that these reduced the risk of VAP compared to placebo or usual care from 24% to about 18% (risk ratio 0.75, 95% confidence intervals (CI) 0.62 to 0.91, I^2^ = 35%) [58]. The second systematic review estimated (with moderate-certainty evidence; 13 randomized concurrent controlled trials; 1206 participants) that these reduced the incidence of VAP compared to placebo or usual care from 26% to about 18% (risk ratio 0.67, 95% CI 0.47 to 0.97; I^2^ = 66%) [59].

With respect to TAP, the third systematic review estimated with TAP combined with PPAP, a strong significant reduction in VAP (odds ratio 0.35; 95% CI 0.29 to 0.41; n = 16) and TAP (including duplex randomized concurrent controlled trials) also estimated a strong significant reduction in VAP (odds ratio 0.56; 95% CI 0.46 to 0.68; n = 17) [60].

The fourth systematic review estimated a significant reduction with respect to TAP combined with PPAP (odds ratio 0.28, 95% CI 0.20 to 0.38; n = 16) and with respect to TAP (including duplex randomized concurrent controlled trials) (odds ratio 0.44, 95% CI 0.31 to 0.63; n = 17) [61].

Finally, the fifth systematic review estimated that, with respect to Topical plus systemic antibiotic prophylaxis, a significant reduction in VAP (risk ratio 0.43, 95% CI 0.35 to 0.53; 17 randomized concurrent controlled trials; 2951 participants; moderate-certainty evidence) and with respect to TAP (including duplex randomized concurrent controlled trials), which may reduce VAP (risk ratio 0.57, 95% CI 0.44 to 0.74; 19 randomized concurrent controlled trials, 2698 participants; low-certainty evidence) [62].

Of particular note in relation to the fifth systematic review is that this review provided illustrative population projections derived (by an unstated method) from the control and intervention groups of the included randomized concurrent controlled trials. For TAP (including duplex randomized concurrent controlled trials), this was 318 per 1000 (i.e., 31.8%) and 137 per 1000 (i.e., 13.7%), respectively. For TAP plus systemic antibiotic prophylaxis, this was 417 per 1000 (i.e., 41.7%) and 238 per 1000 (i.e., 23.8%), respectively. These population projections correspond to the control and intervention group summary VAP incidences derived here.

There are other paradoxical results in relation to both the prevention effect size and the incidences of VAP and other acquired infections such as blood stream infections versus the group mean length of stay among the randomized concurrent controlled trials included within the Cochrane reviews of antimicrobial interventions in this patient population, which implicate spillover contributing to concurrent patients within the ICU and spuriously inflating the apparent effect size of topical antimicrobial based interventions beyond simply being a direct effect limited to only the recipients within the ICU context [95,96].

If either the direct effect of chlorhexidine or the spillover (indirect) effect of TAP was thought to be harmful, it would be ethically difficult to further study them, except as de-adoption studies. Moreover, the logistics of estimating spillover effects are challenging, given that this requires a cluster randomized trial design with thousands of exposed patients in multiple clusters [97].

The SUTVA requires careful consideration in relation to drawing inferences from randomized concurrent controlled trials. Surprisingly, this may have been overlooked by editors of respiratory, sleep, and critical care journals within their recent guidelines on causal inference. These guidelines do not mention or consider SUTVA.

## 5. Conclusions

The possibility that harmful spillover occurred within randomized concurrent controlled trials of topical antimicrobial-based interventions negates the SUTVA and undermines the certainty rating of the VAP data observed in these randomized concurrent controlled trials. The techniques used in systematic reviews typically estimate the intervention effect as an individual-level effect. The use of the caterpillar plot here enables a test of the population effects, which might invalidate the SUTVA.

This spillover from the use of topical chlorhexidine or TAP likely increased the pneumonia incidence above the expected range for the control group incidences among patients receiving mechanical ventilation in the ICU. This spillover, as a population effect, would conflate the summary effect size estimate for these interventions as derived within the systematic reviews. The current guidelines for VAP prevention will likely need to be reviewed, considering this conflation, especially as they relate to the ‘pneumonia zero” initiative.

## Figures and Tables

**Figure 1 jcm-14-06841-f001:**
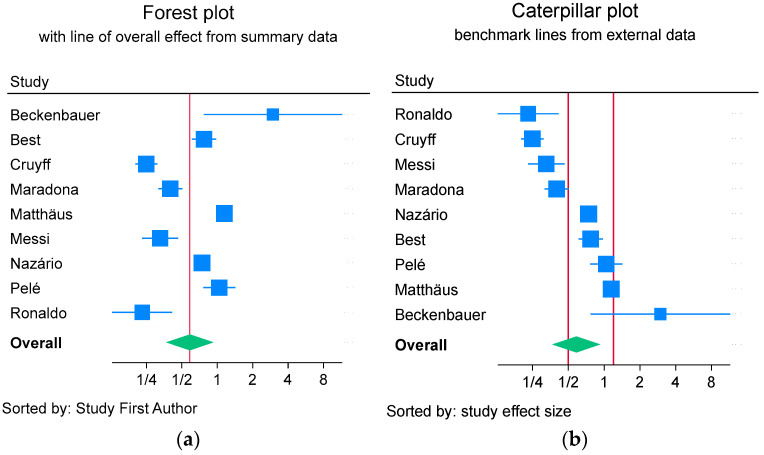
Illustration of a forest plot (**a**) versus caterpillar plot (**b**) with study-specific effect sizes (blue symbols) with 95% confidence limits (blue lines) for individual studies and summary estimates with 95% confidence limits (green diamonds). In the forest plot, the vertical line indicates the summary effect size derived from the included studies, whereas in the caterpillar plot, the vertical lines indicate the benchmark range for the effect size as derived from external sources. In the caterpillar plot analyses here, the effect size is the VAP incidence proportion derived across all control and intervention groups, respectively.

**Figure 2 jcm-14-06841-f002:**
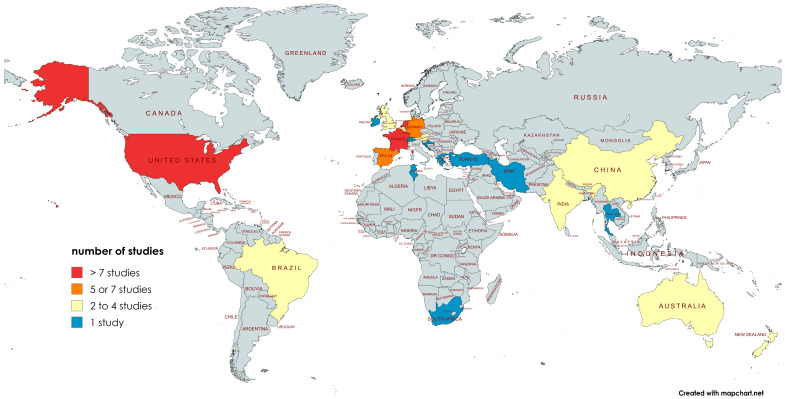
Origins of randomized concurrent controlled trials of topical chlorhexidine or TAP included here.

**Figure 3 jcm-14-06841-f003:**
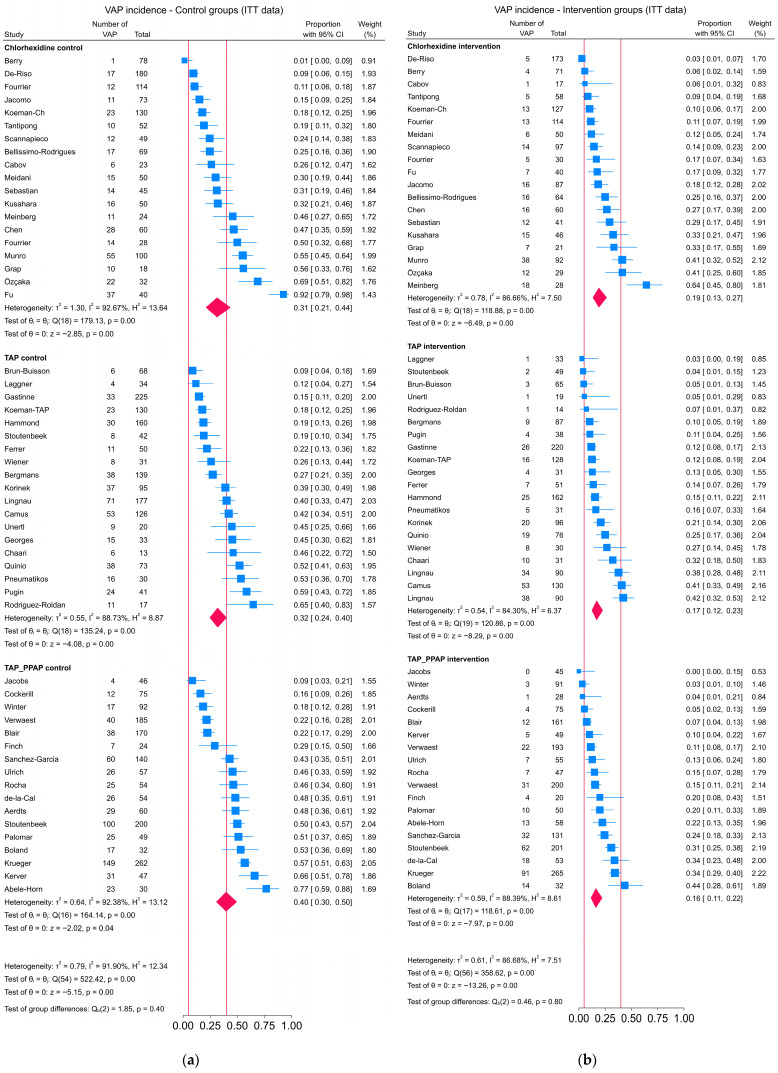
Caterpillar plot with group specific incidence proportions (blue symbols) with 95% confidence limits (blue lines) for individual component groups and summary estimates with 95% confidence limits (red diamonds) of VAP incidence among control (**a**) and intervention (**b**) groups from antimicrobial-based pneumonia prevention randomized concurrent controlled trials drawn from five Cochrane reviews [58,59,60,61,62]. The TAP category includes the duplex randomized concurrent controlled trials. The expert range VAP incidence (5 to 40%) is indicated by the vertical red lines.

**Table 1 jcm-14-06841-t001:** Data extracted from randomized concurrent controlled trials of topical chlorhexidine.

					Patients (n) ^a^			
					Intervention		Control	
Author	Year	Ref.	QS ^b^	Study Intervention ^c^	n	N	n	N
Chlorhexidine studies								
Bellissimo-Rodrigues	2009	[1]	2	Chlx solution	16	64	17	69
Berry	2011	[2]	2	Chlx solution + TB	4	71	1	78
Cabov	2010	[3]	2	Chlx gel	1	17	6	23
Chen	2008	[4]	1	Chlx solution	16	60	28	60
De Riso **^d^**	1996	[5]	2	Chlx solution	5	173	17	180
Fourrier	2000	[6]	2	Chlx gel	5	30	14	28
Fourrier	2005	[7]	2	Chlx gel	13	114	12	114
Fu **^e^**	2019	[8]	1	Chlx solution	7	40	37	40
Grap	2011	[9]	1	Chlx solution	7	21	10	18
Jacomo **^f^**	2011	[10]	2	Chlx solution	16	87	11	73
Koeman-Ch	2006	[11]	2	Chlx gel	13	127	23	130
Kusahara **^f^**	2012	[12]	2	Chlx gel + TB	15	46	16	50
Meidani **^e^**	2018	[13]	1	Chlx solution	6	50	15	50
Meinberg **^g^**	2012	[14]	2	Chlx gel + TB	18	28	11	24
Munro	2009	[15]	1	Chlx solution ± TB	38	92	55	100
Özçaka	2012	[16]	2	Chlx solution	12	29	22	32
Panchabhai **^h^**	2009	[17]	1	Chlx solution (+PP)	14	88	15	83
Scannapieco	2009	[18]	2	Chlx solution + TB	14	97	12	49
Sebastian	2012	[19]	2	Chlx gel	12	41	14	45
Tantipong	2008	[20]	1	Chlx solution + TB	5	58	10	52

Abbreviations: QS = quality score; Chlx = chlorhexidine; TB = toothbrushing; PP = potassium permanganate. (**^a^**) All data are derived as intention-to-treat (ITT) except for Panchabhai [17], which is per-protocol data. (**^b^**) QS = quality score is derived as meeting the majority of quality score criteria in the original Cochrane review. (**^c^**) Study Intervention: Chlx = chlorhexidine; TB = toothbrushing; (+PP = potassium permanganate to control group). (**^d^**) De Rios [5] is a study from a cardiac ICU. (**^e^**) Data for the Fu [8], Meidiani [13] studies derived from Zhao 2020 [59]; all other data from Hua [58]. (**^f^**). Jacomo [10] & Kushara [12] are studies from pediatric ICUs. (**^g^**) Meinberg [14] is from a trauma ICU. (**^h^**) Panchabhai 2009 [17]; reclassified in Zhao’s 2020 systematic review as control group received oral care with potassium permanganate [59]. Only per-protocol data is available for those patients who received the complete protocol. This study is excluded from the current analysis.

**Table 2 jcm-14-06841-t002:** Data extracted from randomized concurrent controlled trials of TAP.

					Patients (n) ^a^			
					Intervention		Control	
Author	Year	Ref.	QS ^b^	Study Intervention ^c^	n	N	n	N
TAP studies								
Bergmans	2001	[21]	2	PGV	9	87	38	139
Brun-Buisson	1989	[22]	2	PNeNa	3	65	6	68
Camus	2005	[23]	1	PT	53	130	53	126
Gastinne	1992	[24]	1	PTA	26	220	33	225
Georges **^d^**	1994	[25]	2	PNeA	4	31	15	33
Koeman-TAP **^e^**	2006	[11]	2	PChlx	16	128	23	130
Korinek	1993	[26]	2	PTAV	20	96	37	95
Pneumatikos **^d^**	2002	[27]	1	PTA	5	31	16	30
Pugin	1991	[28]	2	PNeV	4	38	24	41
Quinio **^d^**	1995	[29]	2	PGA	19	76	38	73
Rodriguez-Roldan	1990	[30]	1	PTNeA	1	14	11	17
Unertl	1987	[31]	2	PGA	1	19	9	20
Wiener	1995	[32]	2	PGNy	8	30	8	31
**Duplex TAP studies**								
Chaari **^d,e^**	2014	[33]	2		10	31	6	13
Ferrer	1994	[34]	2	PTA-Ctx + -Ctx	7	51	11	50
Hammond	1992	[35]	2	PTA-Ctx + -Ctx	25	162	30	160
Laggner	1994	[36]	1	GA + Aug	1	33	4	34
Lingnau **^d,f^**	1997	[37]	2	PTA-Cip + -Cip	38	90	71	177
Lingnau **^d,f^**	1997	[37]	2	PCipA + -Cip	34	90	71	177
Stoutenbeek **^d^**	1996	[38]	2	PTA-C + -C	2	49	8	42

Abbreviations: QS = quality score; (**^a^**) All data are derived as intention-to-treat (ITT). (**^b^**) QS = quality score is derived as meeting the majority of quality score criteria in the original Cochrane review. (**^c^**) Study Intervention: P = polymyxin; T = tobramycin; A = amphotericin; Ne = neomycin, Na = nalidixic acid; Ny = nystatin; no = Norfloxacin; V = vancomycin; G = Gentamicin; Cip = ciprofloxacin; O = Ofloxacin; C = Cefotaxime; Ctx = Ceftriaxone; Ctz = Ceftazidime; Tr = Trimethoprim; Aug = aminopenicillin and clavulanic acid or ‘other appropriate regimens’. (**^d^**) Georges [25], Pneumatikos [27], Quinio [29], Chaari [33], Lingau [37], Stoutenbeek 1996 [38] are studies from trauma ICUs. (**^e^**) Data for the Koeman-TAP [11] and Chaari [33] studies derived from Minozzi [62]; all other data from Liberati [61]. (**^f^**) Lingau [37] provides a three-arm study with the data abstracted from the original source.

**Table 3 jcm-14-06841-t003:** Data extracted from randomized concurrent controlled trials of TAP + PPAP.

					Patients (n) ^a^			
					Intervention		Control	
Author	Year	Ref.	QS ^b^	Study Intervention ^c^	n	N	n	N
TAP + PPAP studies								
Abele-Horn **^d^**	1997	[39]	2	PTA-C	13	58	23	30
Aerdts	1991	[40]	2	PNoA-Ctx	1	28	29	60
Blair	1991	[41]	1	PTA-Ctx	12	161	38	170
Boland **^d^**	1991	[42]	2	PTNy-Ctx	14	32	17	32
Cockerill	1992	[43]	2	NyPG-Ctx	4	75	12	75
de la Cal **^d,e^**	2005	[44]	2	PTA-C	18	53	26	54
Finch	1991	[45]	2	PGA-C	4	20	7	24
Jacobs	1992	[46]	2	PTA-Ctx	0	45	4	46
Kerver	1988	[47]	2	PTA-Ctx	5	49	31	47
Krueger	2002	[48]	2	PG-Cip	91	265	149	262
Palomar	1997	[49]	2	PTA-Ctx	10	50	25	49
Rocha **^d^**	1992	[50]	2	PTA-Ctx	7	47	25	54
Sanchez-Garcia	1998	[51]	2	PTA-Ctx	32	131	60	140
Stoutenbeek **^d^**	2007	[52]	2	PTA-Ctx	62	201	100	200
Ulrich	1989	[53]	1	PNoA-Tr	7	55	26	57
Verwaest **^f^**	1997	[54]	2	PTA-C	22	193	40	185
Verwaest **^f^**	1997	[54]	2	OA-O	31	200	40	185
Winter	1992	[55]	2	PTA-Ctz	3	91	17	92

Abbreviations: QS = quality score; (**^a^**) All data are derived as intention-to-treat (ITT). (**^b^**) QS = quality score is derived as meeting the majority of quality score criteria in the original Cochrane review. (**^c^**) Study Intervention: P = polymyxin; T = tobramycin; A = amphotericin; Ne = neomycin, Na = nalidixic acid; Ny = nystatin; no = Norfloxacin; V = vancomycin; G = Gentamicin; Cip = ciprofloxacin; O = Ofloxacin; C = Cefotaxime; Ctx = Ceftriaxone; Ctz = Ceftazidime; Tr = Trimethoprim; Aug = aminopenicillin and clavulanic acid or ‘other appropriate regimens’. (**^d^**) Abele-Horn [39], Boland [42], De la Cal [44], Rocha [50], and Stoutenbeek 1996 [52] are studies from trauma ICUs. (**^e^**) Data for the De la Cal [44] study derived from Minozzi [62]; all other data from Liberati [61]. (**^f^**) Verwaest [54] is a three-arm study with the data abstracted from the original source.

**Table 4 jcm-14-06841-t004:** Characteristics of studies.

	Chlorhexidine	TAP/Duplex ^a^	TAP + PPAP
Number of studies ^b^	19	20	18
Year (median)	2010	1994	1992
range (min–max)	1996–2019	1987–2014	1988–2007
IQR ^c^	2008–2012	1992–2001	1991–1998
Majority quality score ^d^	13	15	16
Study results			
VAP prevention effect size	0.53	0.46	0.30
95% CI (n)	0.35–0.8	0.33–0.64	0.22–0.42
I^2^	70.7	66.7	65.2
Control group results			
Group VAP—incidence			
• >40%	7	8	11
• ≥5% to ≤40%	11	11	6
• <5%	1	0	0
Total	19	19	17
Group size			
Median	50	50	57
IQR	32–78	31–130	47–140
VAP incidence			
Mean	31	32	40
95% CI ^e^	21–44 (19)	24–40	30–50
I^2^	92.7	88.7	92.4
90% PI ^f^	6–78	11–64	13–74
Intervention group results			
Group VAP—incidence			
• >40%	3	2	1
• ≥5% to ≤40%	15	16	14
• <5%	1	2	3
Total	19	20	18
Group size			
Median	58	58	56
IQR	30–92	31–93	47–161
VAP incidence			
Mean	19	17	16
95% CI ^e^	13–27	12–23	11–22
I^2^	86.7	84.3	88.4
90% PI ^f^	5–53	5–43	5–44

(^a^) A duplex study is defined as a randomized concurrent controlled trial where the control group patients received PPAP. (^b^) Number of eligible studies listed in the original Cochrane review. Note, several randomized concurrent controlled trials had several control or intervention groups. Hence, the number of groups does not equal the number of randomized concurrent controlled trials. Two effect sizes originate from two references each. (^c^) Data is median and inter-quartile range (IQR). (^d^) Majority quality score derived as meeting the majority of quality score criteria in the original Cochrane review. (^e^) 95% CI is 95% confidence interval. (^f^) 90% PI is 90% prediction interval.

**Table 5 jcm-14-06841-t005:** VAP benchmarks.

Citation in Source	Ref.	Original Source
“…with a reported prevalence ranging between 6% and 52%…”	[58], p. 6	[71,72]
“…VAP is a relatively common nosocomial infection in critically ill patients, with pooled incidence from 23.8% to 36.0% in recent systematic reviews…”	[59], p. 9	[73,74]
“…incidence of pneumonia in such patients ranges between 7% and 40%…”	[60], p. 1276	[75]
“The incidence of pneumonia has been reported to vary from 7% to more than 40% in ICU patients”	[61], p. 2	[76,77,78]
“…ventilator-associated pneumonia (VAP) has been estimated to affect 5% to 40% of patients treated with mechanical ventilation for at least 48 h.”	[62], p. 8	[79,80,81]
“…ventilator-associated pneumonia (VAP) continues to complicate the course of 8 to 28%…”	[80], p. 867 (and abstract)	[81,82,83,84,85]

## Data Availability

The data analyzed during the current study are provided in Table 1.

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
