# Peer review of "Testing the Stable Unit Treatment Variance Assumption (SUTVA) Within Cochrane Reviews of Antimicrobial-Based Pneumonia Prevention Interventions Among Mechanically Ventilated Patients Using Caterpillar Plots"

_jcm, 2025, doi:10.3390/jcm14196841_

Round 1

Reviewer 1 Report

Comments and Suggestions for Authors

I have reviewed the manuscript "Testing the stable unit treatment variance assumption (SUTVA) within Cochrane reviews of antimicrobial based pneumonia prevention interventions among mechanically ventilated patients using caterpillar plots" written by James Hurley. 

The overall impression regarding the manuscript is hard to process, difficult to keep focus with all the abbreviations and the information is scarce.

The article is very descriptive regarding the methods and the type of studies analysed and the differences between them, while the aim of the study :"to test the STUVA amongst the findings of Cochrane reviews of antimicrobial based VAP prevention interventions. This requires an arms-based analysis of the VAP incidence proportions within the control and intervention groups of 
these randomized concurrent controlled trials versus the most recently stated expert range."  
The conclusion is somehow supported by the results "The possibility that harmful spillover occurred within randomized concurrent controlled trials of topical antimicrobial based interventions negates the SUTVA assumption and undermines the certainty rating of the VAP data observed in these randomized concurrent controlled trials. This spillover likely increased the pneumonia incidence above the expected range for the control group incidences among patients receiving mechanical ventilation in ICU. This spillover would conflate the summary effect size estimate for these interventions. "

I understand that this manuscript contains a high level of science and impressive statistical analysis, but for the reader is hard to comprehend the aim and the flow of events within the article. Please add information to guide the reader into the subject and make the message of the manuscript understandable.

The whole article is overflowed with data regarding the selection of the randomized controlled trials, number of patients in each study, etc. and there are too few details about the results in the study. Results section should contain only obtained results, not description of what and how was analysed (that belongs to material and method section).

What is the message in Tables 1,2 and 3? If they are only descriptive maybe can be moved to Supplementary files.

In the lines 182- 185 the author described the origin of the studies. Maybe a figure with trial sites on a world map could help visualise easier. 

The limitations of the study are too long.

What is the strength of the study? Maybe discussion could include that positive and valuable idea about the study in the end on discussions section. In my opinion the strength of the current study/review is the unique theme discussed and analysis of already published material  in domain of preventing VAP.

There is a quite large number of self-citation. The reference are very old, only 12 out 99 references are from the last 5 years, References should be numbered as they appear in the text. 

Why does the author consider this manuscript an article rather then a review?

The figure is very detailed. What does the Figure add supplementary to the manuscript with its presence? What is the main idea from the Figure that the reader should comprehend?

Author Response

I have reviewed the manuscript "Testing the stable unit treatment variance assumption (SUTVA) within Cochrane reviews of antimicrobial based pneumonia prevention interventions among mechanically ventilated patients using caterpillar plots" written by James Hurley. 

The overall impression regarding the manuscript is hard to process, difficult to keep focus with all the abbreviations and the information is scarce.

Reply …I have reviewed the manuscript for clarity and minimized the use of abbreviations. The information for all 55 studies is detailed in the tables and figures. There is a table of abbreviations before the references at line 494.

The article is very descriptive regarding the methods and the type of studies analysed and the differences between them, while the aim of the study :"to test the STUVA amongst the findings of Cochrane reviews of antimicrobial based VAP prevention interventions. This requires an arms-based analysis of the VAP incidence proportions within the control and intervention groups of these randomized concurrent controlled trials versus the most recently stated expert range."  The conclusion is somehow supported by the results "The possibility that harmful spillover occurred within randomized concurrent controlled trials of topical antimicrobial based interventions negates the SUTVA assumption and undermines the certainty rating of the VAP data observed in these randomized concurrent controlled trials. This spillover likely increased the pneumonia incidence above the expected range for the control group incidences among patients receiving mechanical ventilation in ICU. This spillover would conflate the summary effect size estimate for these interventions. "

I understand that this manuscript contains a high level of science and impressive statistical analysis, but for the reader is hard to comprehend the aim and the flow of events within the article. Please add information to guide the reader into the subject and make the message of the manuscript understandable.

Reply …I have created a new figure 1 which outlines the key distinguishing points between the forest plot (as commonly used) versus the caterpillar plot (which is the innovation to aid the reader here.

The whole article is overflowed with data regarding the selection of the randomized controlled trials, number of patients in each study, etc. and there are too few details about the results in the study. Results section should contain only obtained results, not description of what and how was analysed (that belongs to material and method section).

Reply …The selection of articles is explained carefully and only once. I am unsure what the reviewer is referring to in relation to material that does not belong in the results. I have reviewed the results section to ensure that any material belonging in the methods is relocated.

What is the message in Tables 1,2 and 3? If they are only descriptive maybe can be moved to Supplementary files.

Reply …The tables 1, 2 & 3 contain essential details about the data being analysed within the caterpillar plots and are required to inform the reader

In the lines 182- 185 the author described the origin of the studies. Maybe a figure with trial sites on a world map could help visualise easier. 

Reply …The simple point here is that the studies originated from various countries. A world map has been added as suggested to help visualise. 

The limitations of the study are too long.

Reply …I tried to include all possible limitations of a novel technique rather than risk missing some. I have added a separate section relating to strengths (see below).

What is the strength of the study? Maybe discussion could include that positive and valuable idea about the study in the end on discussions section. In my opinion the strength of the current study/review is the unique theme discussed and analysis of already published material  in domain of preventing VAP.

Reply … this an analysis using caterpillar plots as a novel method to benchmark systematic review results versus external data. A new section (section 4.5: strengths) is added.

There is a quite large number of self-citation. The reference are very old, only 12 out 99 references are from the last 5 years, References should be numbered as they appear in the text. 

Reply …Unfortunately, the data is from literature which dates mostly from more than 5 years ago. I have minimized self-citation to only those related studies that are essential.

Why does the author consider this manuscript an article rather then a review?

Reply …As noted in the title, this is a test of the SUTVA assumption through an analysis using caterpillar plots as a novel method.

The figure is very detailed. What does the Figure add supplementary to the manuscript with its presence? What is the main idea from the Figure that the reader should comprehend?

Reply … As noted in the title and at lines 206-208, this an analysis using caterpillar plots as a novel method. The figures demonstrate the application of the caterpillar plot. The text and the figure legend has been clarified.

Reviewer 2 Report

Comments and Suggestions for Authors

This manuscript offers a novel and thought-provoking assessment of the Stable Unit Treatment Variable Assumption (SUTVA) in randomized concurrent controlled trials (RCCTs) of antimicrobial interventions for ventilator-associated pneumonia (VAP) prevention. By applying arms-based benchmarking and caterpillar plots to data from Cochrane reviews, the author challenges the validity of traditional effect size estimates and raises the possibility of spillover effects influencing trial results. The work is timely, original, and relevant to both evidence synthesis and critical care practice.

The application of SUTVA concepts and caterpillar plots to this clinical literature is novel and highlights an underappreciated methodological issue. The inclusion of 57 RCCTs across five Cochrane reviews ensures breadth of evidence. The discussion distinguishes clearly between individual-level and population-level effects, and the paradoxical findings between RCCTs and cluster RCTs are well described. Caterpillar plots provide an effective means of visualizing heterogeneity and outliers.

Minor comments

  1. Large cluster RCTs were excluded because of endpoint reporting; however, they provide important counterpoints and should be discussed more fully to avoid bias toward smaller RCCTs.
  2. The discussion should more explicitly address how these findings influence current ICU guidelines and antimicrobial stewardship.
  3. Terminology should be standardized (SUTVA vs. STUVA), and figure legends could include clearer interpretive notes.

Recommendation

Minor Revisions.
The manuscript makes an important contribution by questioning assumptions underlying widely cited trial evidence. 

Author Response

This manuscript offers a novel and thought-provoking assessment of the Stable Unit Treatment Variable Assumption (SUTVA) in randomized concurrent controlled trials (RCCTs) of antimicrobial interventions for ventilator-associated pneumonia (VAP) prevention. By applying arms-based benchmarking and caterpillar plots to data from Cochrane reviews, the author challenges the validity of traditional effect size estimates and raises the possibility of spillover effects influencing trial results. The work is timely, original, and relevant to both evidence synthesis and critical care practice.

Reply …I thank the reviewer for this comment

The application of SUTVA concepts and caterpillar plots to this clinical literature is novel and highlights an underappreciated methodological issue. The inclusion of 57 RCCTs across five Cochrane reviews ensures breadth of evidence. The discussion distinguishes clearly between individual-level and population-level effects, and the paradoxical findings between RCCTs and cluster RCTs are well described. Caterpillar plots provide an effective means of visualizing heterogeneity and outliers.

Reply …I thank the reviewer for this comment

Minor comments

  1. Large cluster RCTs were excluded because of endpoint reporting; however, they provide important counterpoints and should be discussed more fully to avoid bias toward smaller RCCTs.

Reply …The reasons for excluding these studies are detailed at lines 347 to 365. The are fully discussed at lines 381 to 398 with the counterpoints detailed.

  1. The discussion should more explicitly address how these findings influence current ICU guidelines and antimicrobial stewardship.

Reply …The conclusion has been modified and expanded.

  1. Terminology should be standardized (SUTVA vs. STUVA), and figure legends could include clearer interpretive notes.

Reply … Terminology has been standardized (SUTVA) and figure legend clarified.

Recommendation

Minor Revisions.
The manuscript makes an important contribution by questioning assumptions underlying widely cited trial evidence. 

Reply …I thank the reviewer for this comment

Round 2

Reviewer 1 Report

Comments and Suggestions for Authors

The author can remove some of the 8 self-citations if not important or related to current topic.

Author Response

Reviewer 1

Comments: The author can remove some of the 8 self-citations if not important or related to current topic.

Response: I have removed one reference. All other references are essential to the material presented.